# PERSONALIZED RESIDUALS FOR CONCEPT-DRIVEN TEXT-TO-IMAGE GENERATION

## ABSTRACT

We present *personalized residuals* and *localized attention-guided sampling* for efficient concept-driven generation using text-to-image diffusion models. Our method first represents concepts by freezing the weights of a pretrained text-conditioned diffusion model and learning low-rank residuals for a small subset of the model's layers. The residual-based approach then directly enables application of our proposed sampling technique, which applies the learned residuals only in areas where the concept is localized via cross-attention and applies the original diffusion weights in all other regions. Localized sampling therefore combines the learned identity of the concept with the existing generative prior of the underlying diffusion model. We show that personalized residuals effectively capture the identity of a concept in ~3 minutes on a single GPU without the use of regularization images and with fewer parameters than previous models, and localized sampling allows using the original model as strong prior for large parts of the image.

## 1 INTRODUCTION

Large-scale text-to-image diffusion models have demonstrated the ability to generate high-quality images that follow the constraints of the input text Saharia et al. (2022); Rombach et al. (2022); Ramesh et al. (2022). However, these models do not inherently encode any information about the *identity* of a specific concept, thus limiting the control over specifying a particular instance to appear in the generated image. To address this, recent approaches propose techniques to *personalize* these models such that they can generate specific concepts in novel environments and styles.

Given a set of images depicting the desired concept, personalization approaches differ in which parameters they train and whether they are specific to a single concept (i.e., they need to be separately trained for each new concept) or can generalize to new concepts without retraining. To enable personalization of arbitrary concepts, one can finetune the model's parameters Ruiz et al. (2023a) or its inputs Gal et al. (2022) directly such that it can reconstruct the training data. These approaches can be applied to any kind of concepts, but the finetuning needs to be done on a per-concept basis and different parameters need to be stored for each. Other approaches train am encoder specific to a particular domain (e.g., faces) and finetune the diffusion model once to use the encoder's embeddings to reconstruct specific concepts within that domain Xiao et al. (2023); Gal et al. (2023); Ruiz et al. (2023b). The advantage of the latter approach is that it does not require retraining for every concept and can instead be used to instantly generate new concepts from the given domain. However, this approach is limited to a single domain and requires a large dataset to train the encoder.

Our approach follows the former setting, i.e., it finetunes the model's parameters for each concept so that there are no constraints on the domain (see Figure 1 for examples using our proposed method). The main challenges of open-domain approaches is the need for regularization to mitigate forgetting of concepts learned in the model's original training, and the computational overhead in finetuning a new set of parameters for each concept. The most common regularization approach is to use images from the same domain as the target concept with the reference images during the finetuning of parameters. The choice of regularization images affects the quality of the final outputs and, as such, is usually model-, training-, and sometimes even concept-dependent. Finally, to address the large overhead of finetuning a whole new model for each concept, many approaches only finetune a subset of parameters (e.g., attention layers weights Kumari et al. (2023)) or the input to the text-to-image model (e.g., the text embedding representing a specific concept Gal et al. (2022)).

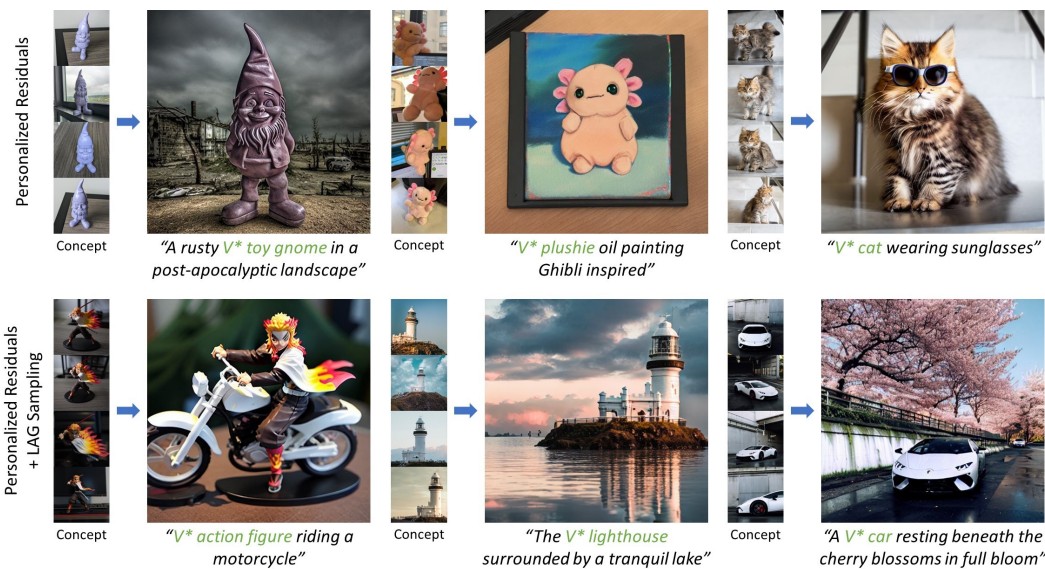

Figure 1: (Top) Given a set of reference images, we learn *personalized residuals* for a subset of a pretrained diffusion model's weights for efficient concept-driven text-to-image generation. (Bottom) The residuals can be combined with our proposed *localized attention-guided (LAG) sampling*, which leverages the cross-attention maps from the diffusion models to localize the application of the residuals and uses the original, unchanged, diffusion model for generating everything else.

Our approach further reduces the number of learnable parameters and does not rely on regularization images. While most approaches focus on finetuning the key and value weights of the cross-attention layers, we instead predict a low-rank residual to the weights of the output projection conv layer after each cross-attention layer. This allows us to finetune even fewer parameters (about ∼0.1% of the base model) than previous approaches. Furthermore, we find that this approach does not require any regularization images which makes our approach both simpler, since we do not need to find appropriate strategies to obtain regularization images, and faster, since we do not need additional training iterations for learning from the regularization images.

Additionally, many personalization approaches struggle to render specific backgrounds or add new objects often due to some degree of overfitting to the target concept. For these scenarios, we propose a novel localized attention-guided (LAG) sampling scheme, which allows us to use the finetuned residuals with the original model to generate the target concept and the rest of the image, respectively. To achieve this, we use the attention maps from the cross-attention layers of the diffusion model at each timestep to predict the location of the concept in the generated image and then apply the features, produced using the personalized residuals, only in the predicted region such that the rest of the image (e.g., background and other objects) is generated by the original model. Thus, we ensure that we do not lose the capability of generating specific backgrounds or unrelated objects due to overfitting. Furthermore, this sampling approach does not require any additional training or data, and does not increase sampling time as no additional model evaluations are needed.

We evaluate our approach and sampling technique on the CustomConcept101 dataset Kumari et al. (2023), which was specifically designed to evaluate personalization approaches. We use CLIP and DINO scores to evaluate the text-image alignment (i.e., how well the personalized model can generate the concept in novel scenes and environments) and identity preservation of the personalized model (i.e., how well it can generate the desired concept). We also perform a user study to evaluate human preference for text-image alignment and identity preservation. Our results show that our model performs on par or better compared to current state-of-the-art baselines while using significantly fewer parameters, not relying on regularization images, and being faster to train.

To summarize, our key contributions are a novel low-rank personalization approach for text-to-image diffusion models that works for arbitrary domains and concepts, uses fewer parameters than previous approaches, does not rely on regularization images and is, therefore, faster and simpler to train. We also introduce a novel *localized attention-guided (LAG) sampling* approach that allows us

to flexibly combine the original pretrained and the finetuned model on the fly to generate different parts of the image, without increasing the sampling time and without requiring additional training or user inputs. Our user study and quantitative evaluations show that our method performs comparably or better than other baselines, and our proposed sampling approach can address challenges with certain types of recontextualization scenarios, such as background changes.

## 2 RELATED WORK

### 2.1 PERSONALIZATION OF TEXT-TO-IMAGE MODELS

The task of text-to-image personalization was proposed by Gal et al. (2022), where a few example images of the given concept are used to finetune a "personalized" token embedding while all other parameters of the model frozen. Instead of trying to find an embedding within the existing text conditioning space to represent a concept, DreamBooth Ruiz et al. (2023a) finetunes the diffusion model's parameters to directly inject the concept into the learned prior, leading to better performance. Custom Diffusion Kumari et al. (2023) only finetunes the cross-attention weights in addition to the token embedding to achieve more efficient personalization compared to DreamBooth. Based on these works, other aim to improve the performance and efficiency of personalizing text-to-image models through approaches such as, but not limited to, learning multiple personalized tokens Dong et al. (2022); Hertz et al. (2022), imposing constraints on the trainable parameters (e.g., key-locking Tewel et al. (2023), orthogonality Qiu et al. (2023), low-rank Smith et al. (2023), singular values only Han et al. (2023)), training hypernetworks and domain-specific encoders Ruiz et al. (2023b); Xiao et al. (2023); Li et al. (2023); Gal et al. (2023), and injecting of visual features Wei et al. (2023); Hao et al. (2023); Xiao et al. (2023).

### 2.2 ATTENTION-GUIDED TEXT-TO-IMAGE SYNTHESIS

Attention layers Vaswani et al. (2017) have been shown to play an important role in the success of text-conditioned image synthesis using diffusion models. Recent works propose to manipulate attention maps from these layers for guided synthesis and editing. Balaji et al. (2022) enable a "paint-by-words" synthesis approach conditioned on a user-provided layout by guiding the localization of objects via cross-attention manipulation. Given an existing image and a prompt that describes the image, Prompt2Prompt Hertz et al. (2022) edits an image by swapping out the cross-attention map corresponding to the target with one corresponding to the desired edit. Similarly, InstructPix2Pix Brooks et al. (2023) performs edits on existing images albeit through instructions and modifications within self-attention layers.

## 3 APPROACH

Our method consists of two components: 1) *Personalized residuals*, which encode the identity of a given concept through a set of learned offsets applied to a subset of weights within a pretrained text-to-image diffusion model, and 2) *Localized attention-guided (LAG) sampling*, which leverages attention maps to localize where the residuals are applied, essentially allowing a single image to be efficiently generated by leveraging both the base diffusion model and the personalized residuals.

### 3.1 PRELIMINARIES

**Diffusion models.** Diffusion models Ho et al. (2020) consist of a fixed forward noising process that gradually adds noise to an image, and a learned denoising process that iteratively removes noise to produce a valid image. The denoising process is learned through a U-Net Ronneberger et al. (2015) $\epsilon_\theta$, parameterized by $\theta$, and is conditioned on an image $x_t$ noised to timestep $t$, and $t$ itself. Text guidance can be incorporated through conditioning on embeddings $c = \tau(y)$ of input prompts $y$ from a text encoder $\tau$, such as CLIP Radford et al. (2021).

In this work, we leverage Stable Diffusion, a text-conditioned latent diffusion model (LDM) Rombach et al. (2022). An LDM is a variant of a diffusion model that operates in the latent space of a variational autoencoder Kingma & Welling (2014). The encoder $\mathcal{E}$ embeds an input image $x$ into a latent representation $z = \mathcal{E}(x)$ and a decoder $\mathcal{D}$ maps $z$ back into pixel space $x' = \mathcal{D}(z)$. The diffusion portion of LDM operates on $z$ and is trained using the following objective:

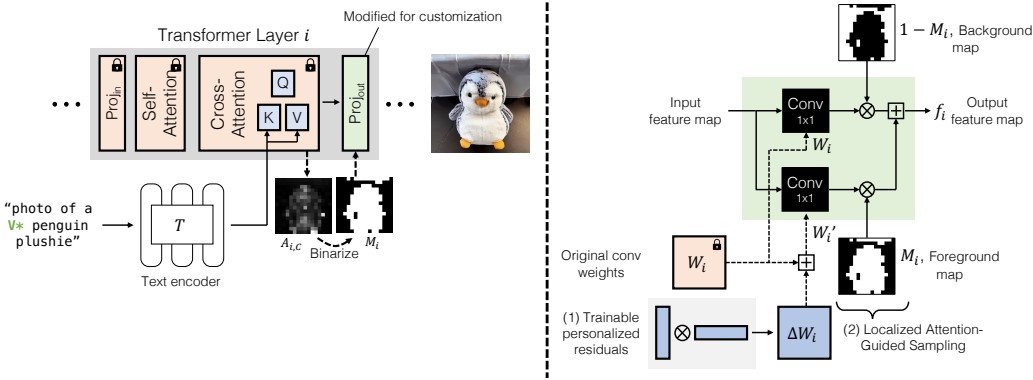

Figure 2: Overview of our proposed work. (1) *Personalized residuals*: We learn low-rank residuals for the output projection layer within each transformer block in the diffusion model. The residuals contain relatively few parameters, are fast to train, and do not require any regularization images during training. (2) *Localized attention-guided sampling*: We optionally apply the personalized residuals only in the areas that the cross-attention layers have localized the concept via predicted attention maps. Thus, we can combine the newly learned concept with the original generative prior of the base diffusion model within a single image.

$$\mathcal{L}_{\text{LDM}} = \mathbb{E}_{z \sim \mathcal{E}(x), y, \epsilon \sim \mathcal{N}(0,1), t}\left[\|\epsilon - \epsilon_\theta\left(z_t, t, \tau(y)\right)\|_2^2\right]. \tag{1}$$

**Low rank adaptation (LoRA).** Low rank adaptation (LoRA) Hu et al. (2021) is an efficient method originally proposed for updating large language models through learned residuals instead of directly finetuning their parameters. For a given layer of the pretrained model with weight matrix $W_0 \in \mathbb{R}^{m \times n}$, LoRA learns two matrices $A$ and $B$ whose product forms a residual $\Delta W = AB \in \mathbb{R}^{m \times n}$, where $A \in \mathbb{R}^{m \times r}$, $B \in \mathbb{R}^{r \times n}$, and $r \ll \min(m, n)$ is the rank. The updated weight matrix is then defined as $W' = W_0 + \Delta W$. With small values of $r$, LoRA has been shown to significantly reduce the number of learnable parameters while retaining or even improving performance.

### 3.2 LEARNING RESIDUALS FOR CAPTURING IDENTITY

The goal of personalizing text-to-image models is to faithfully capture the identity of a target concept while simultaneously avoiding overfitting so that the concept can be recontextualized into new settings and configurations. Since concepts are often learned using only a few reference images, directly finetuning the weights of a very large generative model can easily lead to overfitting and/or overwriting unnecessary parts of the learned language prior. Instead we propose to use a LoRA-based approach, to learn low-rank offsets for a small subset of the diffusion model weights which will represent the target concept while better preserving the existing prior.

The diffusion model contains multiple transformer blocks, which consist of self- and cross-attention layers Vaswani et al. (2017) with a $1 \times 1$ conv projection layer on either end (see Figure 2). While several approaches primarily target the cross-attention layers due to their learning of relationships between text and images, we choose to learn offsets for the output projection conv layers because these localized operations can capture finer details than the global operations of cross-attention.

We illustrate the process of learning personalized residuals in Figure 2. Given a pretrained text-to-image diffusion model containing $L$ transformer blocks, we learn $\Delta W_i = A_i B_i \in \mathbb{R}^{m_i \times m_i}$ for the output projection layer $l_{\text{proj\_out,i}}$ with weight matrix $W_i \in \mathbb{R}^{m_i \times m_i \times 1}$ within each transformer block $i$, where $A_i \in \mathbb{R}^{m_i \times r_i}$ and $B_i \in \mathbb{R}^{r_i \times m_i}$. We reshape the residual such that $\Delta W_i \in \mathbb{R}^{m_i \times m_i \times 1}$ and add to the original weights $W_i$ to produce $W_i' = W_i + \Delta W_i$. The $\Delta W_i$'s are updated using the original diffusion objective in Equation (1).

Similar to other works, we associate the concept with a unique identifier token (e.g., V*), which is initialized using a rarely occurring token embedding. During training, we use the unique token and macro class of the concept in a fixed template for the prompt associated with each reference image (e.g., "a photo of a V* macro class"). Personalization approaches that involve direct updates

to the diffusion model's weights are susceptible to overwriting parts of the existing generative prior with the new concept and thus explicitly require "prior preservation" through regularization images during training Ruiz et al. (2023a); Kumari et al. (2023). Since our method does not directly update the diffusion model, we avoid this issue entirely and eliminate the burden on the user to determine an effective set of regularization images, which is not always straightforward. Additionally, the low-rank constraint on the residuals reduces the number of trainable parameters, making our method a simpler and more efficient approach for personalization.

### 3.3 LOCALIZED ATTENTION-GUIDED SAMPLING

With our residual-based personalization approach, we have additional flexibility in how the offsets are applied at inference. We introduce a new *localized attention-guided* (LAG) sampling method to better combine a newly learned concept with the original generative prior of the diffusion model. As shown in Figure 2, within every transformer block of the diffusion model is a cross-attention layer, which aims to learn the correspondence between text tokens and image regions. Each cross-attention layer computes attention maps $A_{y_i}$ for each token $y_i$ in the prompt, indicating where the token will affect the generated image. The attention maps are produced using the following equation:

$$A(Q, K) = \text{softmax}\Big(\frac{QK^\top}{\sqrt{d_k}}\Big), \tag{2}$$

where $Q = W^Q x$ is the query, $K = W^K y$ is the key, and $d_k$ is the dimension of the query and key.

Given the indices $\mathcal{C}$ of the unique identifier and macro class tokens specifying the concept (e.g., "$\mathbb{V}\star$" and "dog"), we sum the values of the corresponding attention maps $A_{i,\mathcal{C}} = \sum_{j \in \mathcal{C}} A_j$ in transformer block $i$, and then binarize using its median value to get $M_i = \text{binarize}(A_{i,\mathcal{C}})$. Finally, we compute the output feature $\hat{f}_i$ of each transformer block $i$ as:

$$\hat{f}_i = (1 - M_i) \otimes f_i + M_i \otimes f_i', \tag{3}$$

where $f_i = W_i x$ is the feature produced using the original conv weight $W_i$, and $f_i' = W_i' x$ is the feature produced using the updated weight from the personalized residual $W_i' = W_i + \Delta W_i$. Thus, the identity represented through the personalized residuals is only being applied in the regions corresponding to the target concept, and the remaining regions are generated by the original diffusion model. The proposed LAG sampling technique is visualized in Figure 4.

LAG sampling can be beneficial in scenarios where the learned residuals overfit to the reference images and have not effectively disentangled the target concept from the background, which can occur as a consequence of ambiguities of the target concept given the reference images or model biases (e.g., furniture is often photographed indoors). By leveraging the attention maps from the tokens denoting the concept, we can localize the residuals so that they do not affect the background, which can instead be generated using the base model.

## 4 EXPERIMENTS

In this section, we describe our experimental setup and evaluation protocols, and visualize examples using the proposed personalized residuals with and without localized attention-guided sampling.

### 4.1 TRAINING DETAILS

We build upon Stable Diffusion v1.4 Rombach et al. (2022). For each transformer block $i$, we compute the rank $r_i$ for its output projection convolution layer with weight matrix $W_i \in \mathbb{R}^{m_i \times m_i \times 1}$ as $r_i = 0.05 m_i$, totalling 1.2M trainable parameters ($\sim 0.1\%$ of Stable Diffusion). Each of the low-rank matrices are randomly initialized. We train our method for 150 iterations with a batch size of 4 and learning rate of 1.0e-3 on 1 A100 GPU ($\sim 3$ minutes) across all experiments.

Table 1: Quantitative evaluations for text and image alignment using the similarity of CLIP and DINO features. We report the number of parameters for each method in addition to scores from the base Stable Diffusion model, which is not trained for personalization, for reference.

| Method | # params | CLIP text | CLIP image | DINO image |
|---|---|---|---|---|
| Textual Inversion | 768 | 0.6150 | 0.7259 | 0.4700 |
| DreamBooth | 983M | 0.7536 | 0.7424 | 0.5212 |
| Custom Diffusion | 19M | **0.7664** | 0.7074 | 0.4669 |
| Ours | 1.2M | 0.7193 | **0.7594** | **0.5671** |
| Ours w/ LAG sampling | 1.2M | 0.7220 | 0.7424 | 0.5411 |
| Stable Diffusion | 983M | 0.8126 | 0.6207 | 0.2920 |

## 4.2 BASELINES

We focus on comparisons to published and open-domain (i.e., does not require encoders limited to a single given domain) approaches with publicly available code. Specifically, we compare our method against three baselines: Textual Inversion Gal et al. (2022), DreamBooth Ruiz et al. (2023a), and Custom Diffusion Kumari et al. (2023). Textual Inversion freezes the entire diffusion model and optimizes only the unique identifier token V* for each concept. DreamBooth finetunes the entire diffusion model using the reference images and a set of regularization images, which are generated within the same domain as the target concept using the original model. Custom Diffusion finetunes only the key and value weights of the cross-attention layers in addition to the identifier token embedding, and uses a set of real regularization images sampled from LAION-400M Schuhmann et al. (2021). While DreamBooth was originally proposed using Imagen Saharia et al. (2022), we use an open-source version built on Stable Diffusion[1].

We use the recommended settings described by each paper. For Textual Inversion, which initializes the identifier token embedding to a single word that best represents the concept, we use our best discretion to pick a word most similar to the macro class given by CustomConcept101.

## 4.3 EVALUATION METRICS

Following the protocol described in Kumari et al. (2023), we leverage the CustomConcept101 dataset, consisting of 101 concepts across 16 broader categories. For every concept we generate 50 samples for each of the 20 prompts given by the dataset. We use DDIM sampling Song et al. (2020) with $N = 50$ steps, $\eta = 0.0$, and a guidance scale of 6.0 for all methods. We set the same random seed for sampling across each method so that the "choice" of starting noise does not impact the results. Results of our method with LAG sampling are explicitly labeled as such.

We evaluate each method for text alignment and image alignment. *Text alignment* is measured as the similarity between the CLIP Radford et al. (2021) text feature of the input prompt and the CLIP image feature of the resulting generated image. *Image alignment* is measured as the similarity between image features from either CLIP or DINO Caron et al. (2021) of the reference images and corresponding generated images.

Additionally, we evaluate both text and image alignment using human evaluations through user studies on Amazon Mechanical Turk (AMT). For each text alignment case, we display a text prompt and a pair of corresponding generated images, and ask users *"Which image is more consistent with the given text prompt?"*. For each image alignment case, we display 3 reference images for a concept and a pair of corresponding generated images, and ask *"Which image better preserves the identity of the subject in the provided reference images?"*. For both studies, each pair of images contains one from {Textual Inversion, DreamBooth, Custom Diffusion, Ours w/ LAG sampling} and one from ours with normal DDIM sampling. Users can select either image or neither (*"Not sure"*).

## 4.4 RESULTS

We visualize samples generated by each method for various types of prompts in Figure 3. Textual Inversion fails to reliably capture the concept's identity and/or the prompt whereas all other

---

[1]https://github.com/XavierXiao/Dreambooth-Stable-Diffusion

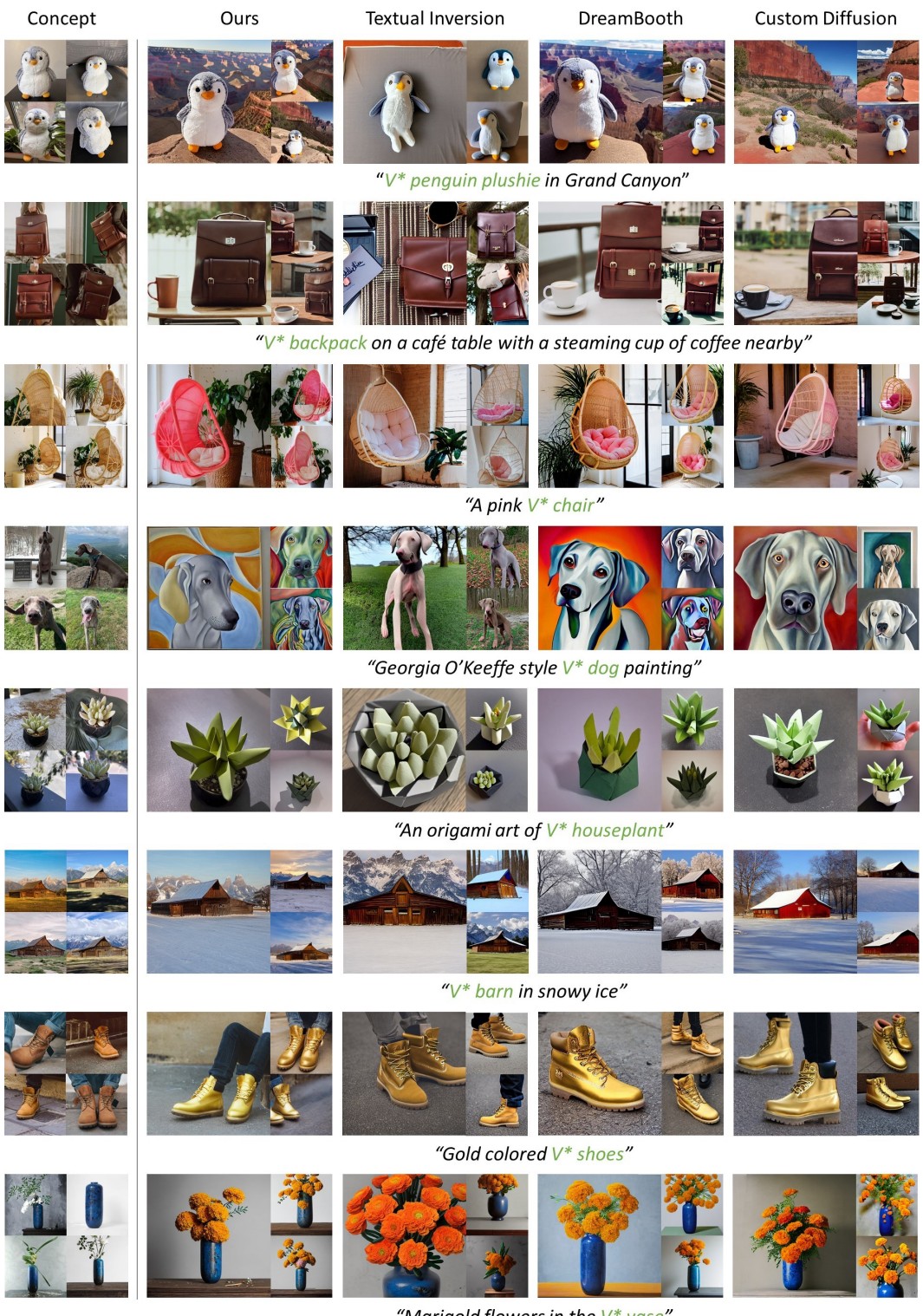

Figure 3: Qualitative comparison of our proposed approach with the baselines.

Table 2: Human preference evaluations for text and image alignment through Amazon Mechanical Turk. We perform bootstrap resampling over the 1000 responses collected for each task.

| Ours vs. | Textual Inversion | DreamBooth | Custom Diffusion | Ours w/ LAG sampling |
|---|---|---|---|---|
| Text | **81.85** ±4.15% | 41.34 ±5.08% | **50.99** ±5.46% | **58.57** ±6.34% |
| Image | **61.96** ±4.76% | **51.33** ±4.65% | **63.27** ±5.59% | 26.26 ±4.91% |

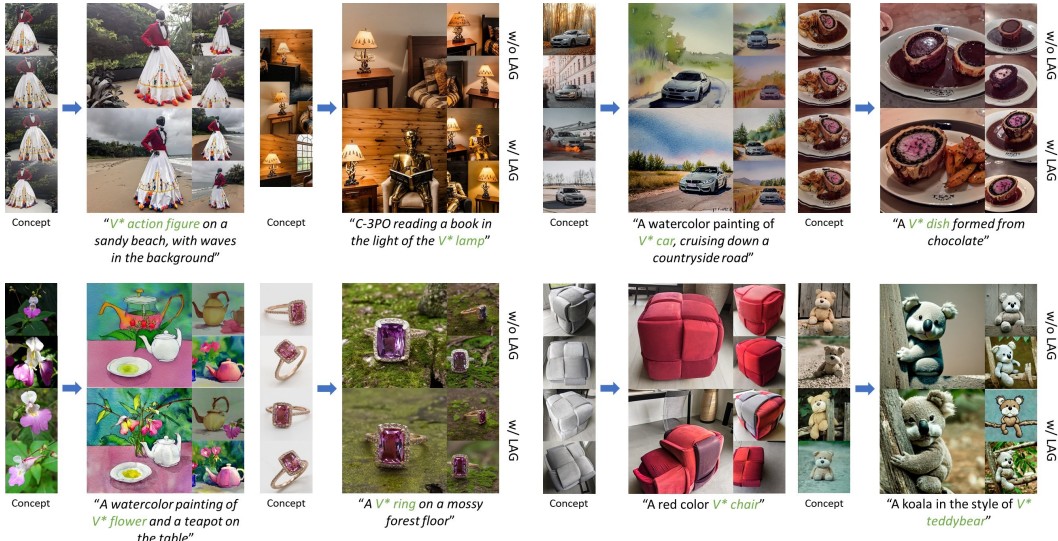

(a) Examples where LAG produces results that are better aligned with the concept and prompt.

(b) Examples where normal sampling produces results that are better aligned with the concept and prompt.

Figure 4: Comparison of image generated with and without LAG sampling. We use the same starting noise map to generate corresponding pairs of images to directly visualize how LAG sampling affects the output image.

methods, including ours, are able to better preserve the concept's identity while also adhering to the prompt. We highlight that our method is able to achieve these results while having significantly fewer parameters, requiring less training time, and not leveraging regularization images compared to DreamBooth and Custom Diffusion.

We compare examples using our proposed personalized residuals with and without localized attention-guided sampling in Figure 4. We illustrate how LAG sampling affects the output image by using the same starting noise map $z_T$ to sample each pair of {w/o LAG, w/ LAG} images. Figure 4a highlights scenarios where LAG sampling generally tends to perform better than normal sampling (e.g., changing scenery/background, adding objects), and Figure 4b highlights scenarios where normal sampling performs better (e.g., varying artistic style, changing attributes/properties). Artistic styles can benefit from LAG sampling when the texture of the concept is similar to the desired style (e.g., watercolor flower in Figure 4a) otherwise the localization causes the texture of the concept to mismatch the rest of the image (e.g., watercolor car in Figure 4b).

Quantitative evaluations for text and image alignment using CLIP and DINO are shown in Table 1. We include results using the original Stable Diffusion model, which has no notion of any of the concepts, for reference. We show that our method performs similarly with and without LAG sampling averaged across the whole dataset, demonstrating higher image alignment and slightly lower text alignment than the more computationally-heavy baselines.

However, as seen by the results of 1000 responses collected through AMT user studies for both text and image alignment in Table 2, we show that the CLIP text alignment scores do not necessarily

Table 3: We evaluate our method using two different targets for the residuals and altering various training settings.

| Method | CLIP text | CLIP image | DINO image |
|---|---|---|---|
| KV weights | 0.7172 | 0.7508 | 0.5353 |
| $l_{\mathrm{proj\_in}}$ weights | 0.7136 | 0.7460 | 0.5333 |
| w/o macro class | 0.6605 | 0.6521 | 0.3798 |
| w/ reg images | 0.7204 | 0.6771 | 0.3830 |
| Update token embedding | 0.6673 | 0.8000 | 0.6194 |
| Ours | 0.7193 | 0.7594 | 0.5671 |

correlate to human preference. We observe that our method performs similarly to Custom Diffusion for text alignment, which was assigned the highest CLIP text score, and outperforms all baselines for image alignment. We also compare our method with and without LAG sampling in the user studies and show that LAG is preferred for image alignment but not text alignment. This may be due to the number of prompts in the dataset for which LAG is generally worse for; the car in the watercolor painting using LAG in Figure 4b technically preserves the original attributes of the car better, but the painting generated without LAG adheres to the specified style better.

We also train and evaluate our method using CLIP similarity to select the "most representative" macro class among the 117k nouns in WordNet Miller (1995) for each concept. In Table 4, we show that using the WordNet macro class leads to further improvements in image alignment while decreasing text alignment, the latter of which may not necessarily reflect human preference as previously demonstrated. See Appendix B for additional discussions.

**Ablation studies.** We perform ablation studies on changing the targets for where the residuals are applied, removing the macro class from the prompt, including regularization images (sampled from LAION) during training, updating the concept identifier token embedding $\mathtt{V}*$, and varying the rank of the residuals. Results are shown in Table 3 (see Table 5 for results on changing the rank).

We show that changing where the residuals are applied to either the key and value weights of the cross-attention layers (like Custom Diffusion) or the input projection conv layer (rather than the output) slightly decreases the scores across all three metrics compared to our proposed approach. We hypothesize that the output projection layer achieves noticeably higher identity preservation because it refines the feature map at the end of each block.

Omitting the macro class leads to significant drops across all metrics, demonstrating that the additional information is useful to our method for knowing what within the reference images is important to model. Similar to the effect of using regularization images for DreamBooth and Custom Diffusion, regularization images slightly improves text alignment but decreases image alignment. On the other hand, updating the token embedding for $\mathtt{V}*$ leads to overfitting as shown by the increase in image alignment and decrease in text alignment.

## 5 CONCLUSION

We introduce personalized residuals, a method for concept-driven synthesis using text-to-image diffusion models. Previous approaches to personalization are often slow to train, have high computational demands, require regularization images, and/or have difficulty recontextualizing the target concept. Through our proposed LoRA-based approach that learns a small set of residuals to represent the identity of a concept, we reduce the number of learnable parameters and training time and remove the reliance on domain regularization while maintaining flexibility with editing. We also introduce localized attention-guided sampling which applies the personalized residuals only in regions where the concept is localized via the cross-attention mechanism. We evaluate our method across several metrics to show that we are able to efficiently enable personalization.

That said, we show that localized sampling is not always the best choice (e.g., changing the color of a concept) and relies on the cross-attention layers to produce high-quality attention maps, which is not always the case. Our approach can be sensitive to the choice of macro class and inherits the pretrained model's biases and limitations, such as mixing up the relationship between attributes in the prompt. Finally, we leave multi-concept generation through LAG sampling as future work.

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

## A  ADDITIONAL EXPERIMENTAL RESULTS

We provide additional qualitative examples generated using our proposed personalized residuals with and without localized attention-guided sampling in Figures 5 and 6. We also plot CLIP/DINO image alignment scores against CLIP text scores, averaged across concepts within the the 16 categories of CustomConcept101, for each method from Section 4.

## B  EFFECT OF MACRO CLASS CHOICE

Table 4: We compute the nearest neighbor (NN) in CLIP embedding space for each concept among all WordNet nouns. We compare our method using different combinations of macro classes (from CustomConcept101 or WordNet NN) during training and sampling.

| Macro class choice | | CLIP text | CLIP image | DINO image |
|---|---|---|---|---|
| Training | Sampling | | | |
| CustomConcept101 | CustomConcept101 | 0.7193 | 0.7594 | 0.5671 |
| | WordNet NN | 0.7155 | 0.7594 | 0.5671 |
| WordNet NN | CustomConcept101 | 0.6626 | 0.7798 | 0.5904 |
| | WordNet NN | 0.6869 | 0.7798 | 0.5904 |

For each concept in CustomConcept101, we compute the mean CLIP image embedding of all the reference images and calculate the cosine similarity against the CLIP text embedding for each of the 117k nouns within WordNet. We train our method and/or sample using the WordNet noun with the highest similarity and compare with using the provided macro class from CustomConcept101 during training and/or sampling in Table 4. We observe that using the WordNet nearest neighbor as the macro class leads to higher image alignment and lower text alignment compared to the CustomConcept101-provided macro class.

Selecting the "best" macro class for concepts can be challenging and given that the macro class can lead to noticeable changes in alignment metrics, an automatic heuristic for choosing a suitable macro class would be helpful to users. We leave the designing of such a heuristic as future work.

## C  ABLATION STUDY: RANK VALUE

Table 5: Quantitative evaluations for varying the rank of the learned residuals. $m_i$ is the dimension of the weight of the projection layer in transformer block $i$.

| Rank | CLIP text | CLIP image | DINO image |
|---|---|---|---|
| 1 | 0.7398 | 0.6809 | 0.4148 |
| 8 | 0.7054 | 0.7402 | 0.5239 |
| 16 | 0.6926 | 0.7573 | 0.5513 |
| 32 | 0.6832 | 0.7701 | 0.5713 |
| 64 | 0.6704 | 0.7798 | 0.5865 |
| 128 | 0.6544 | 0.7938 | 0.6053 |
| $0.025m_i$ | 0.6889 | 0.7622 | 0.5595 |
| Ours ($0.05m_i$) | 0.7193 | 0.7594 | 0.5671 |

We evaluate different values for the rank of the learned residuals in Table 5 and observe that text alignment is inversely proportional to the rank and image alignment is directly proportional. Since the dimensions of the conv weight matrix varies across the transformer blocks within the U-Net, we believe that calculating the rank with respect to the dimensions is the better approach over setting a fixed value across all layers, which is empirically validated by the results with our proposed formula achieving a better balance of image and text alignment.

Concept          Ours          Ours w/ LAG sampling

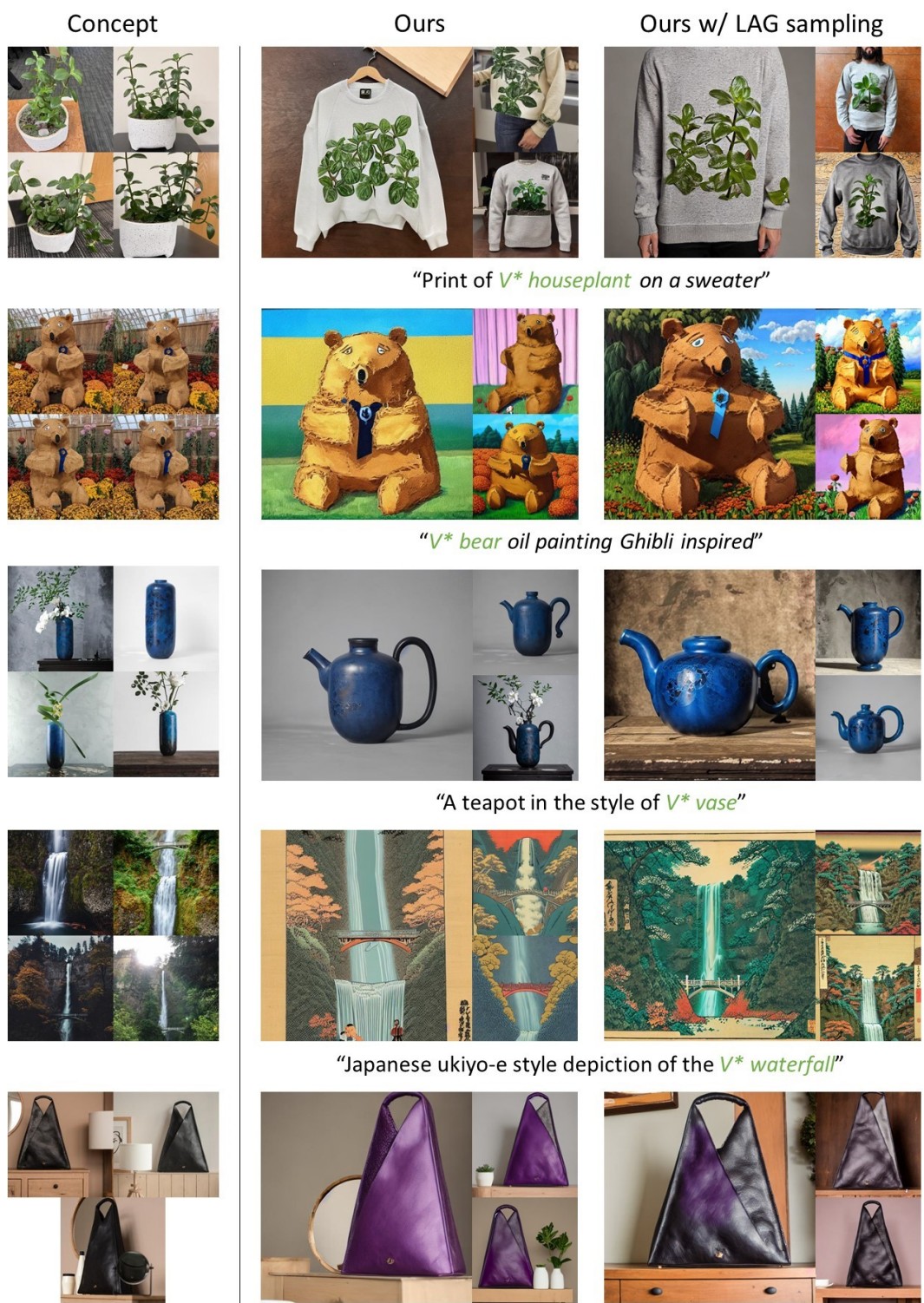

Figure 5: Samples generated using personalized residuals with and without LAG sampling.

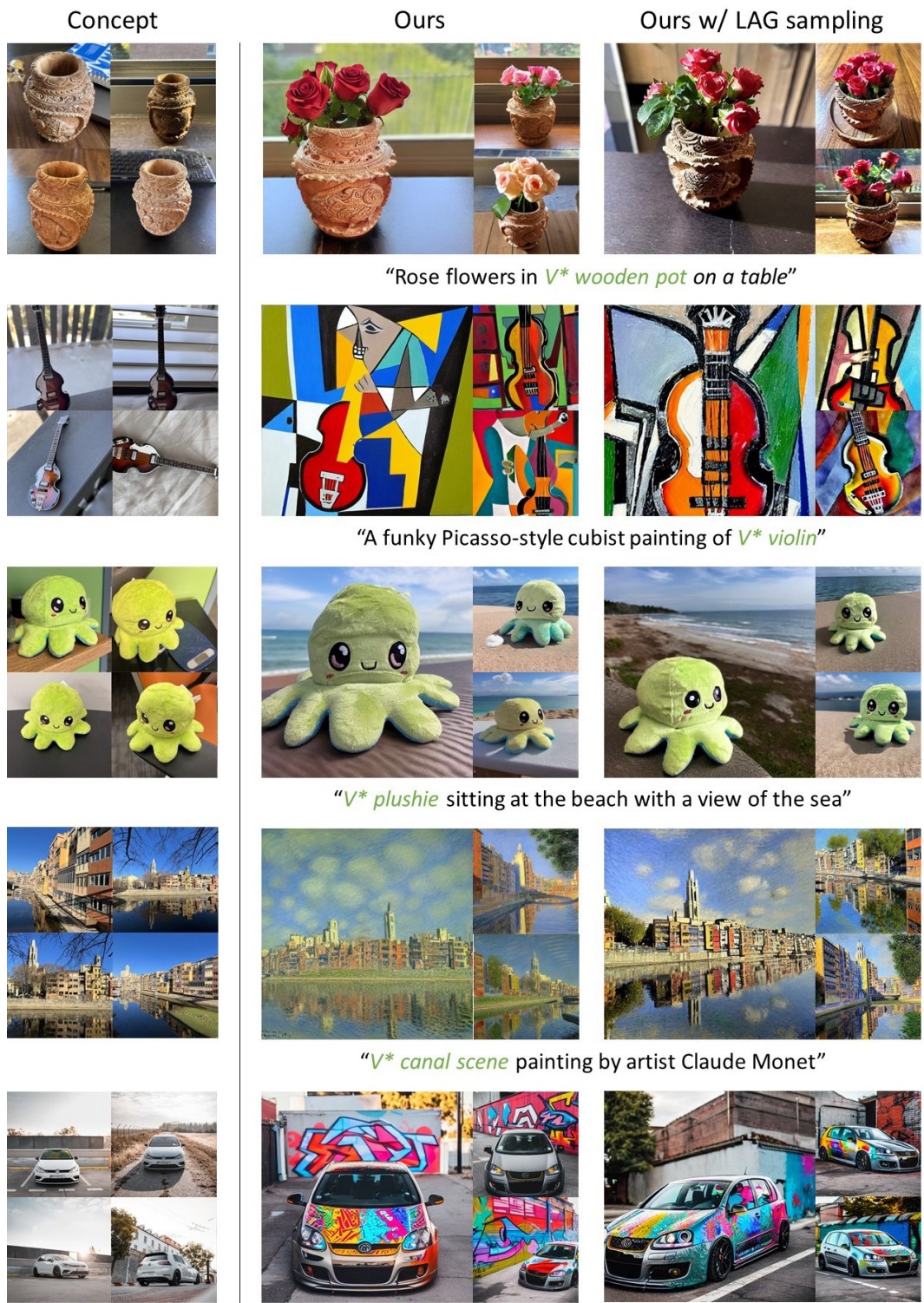

Figure 6: Samples generated using personalized residuals with and without LAG sampling.

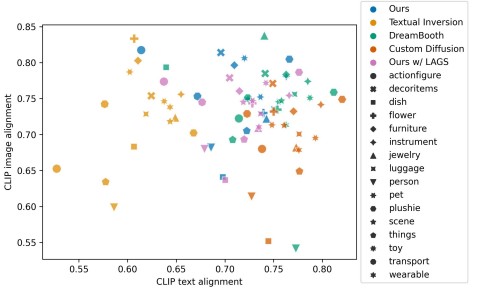

(a) Plot of CLIP image alignment vs. CLIP text alignment.

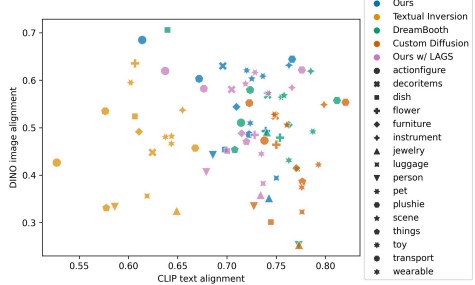

(b) Plot of DINO image alignment vs. CLIP text alignment.

Figure 7: For each method, we plot the either CLIP or DINO image alignment scores against CLIP text alignment scores averaged across the concepts within each of the 16 categories of CustomConcept101.

