# OpenReview forum: "Personalized Residuals for Concept-Driven Text-to-Image Generation"
_ICLR.cc/2024/Conference — ICLR 2024 Conference Withdrawn Submission_

### Official Review · Reviewer_szNh · 2023-10-24

**Soundness:** 3 good
**Presentation:** 3 good
**Contribution:** 2 fair
**Rating:** 5
**Confidence:** 4

**Summary:**

The study introduces a method for efficient concept-driven image generation using text-to-image diffusion models. It represents concepts by adjusting a pretrained model's weights and utilizes localized attention-guided sampling. This approach captures concept identity quickly on a single GPU, using fewer parameters than previous models, while maintaining the generative potential of the original model for most of the image.

**Strengths:**

1. Efficiency: The method offers a more efficient approach to personalized image generation, using fewer parameters and avoiding the need for regularization images, resulting in faster and simpler training.
2. Domain Flexibility: It can be applied to arbitrary domains and concepts, making it versatile and adaptable to a wide range of image generation tasks.
3. Improved Sampling: The localized attention-guided (LAG) sampling approach enhances the generation process by focusing on areas where the concept is localized, combining concept identity with the strengths of the original model for better results.

**Weaknesses:**

1. Limited Novelty: The paper is criticized for its limited novelty compared to existing work, suggesting that it may not significantly advance the state-of-the-art in the field.
2. Lack of Quality Improvement: Reviewers note that the method does not substantially improve the quality of generated images when compared to existing approaches, as evidenced by Figure 3 and Table 2.
3. Insufficient Baseline Comparisons: The paper is faulted for not including thorough discussions or experiments comparing its method against strong baselines, such as IP-adapter or ControlNet reference-only mode, which makes it challenging to assess its relative performance.
4. Control Support Unclear: There is ambiguity regarding whether this method supports adding control, like IP-adapter also supports ControlNet, as this is not adequately addressed or demonstrated in the paper.

These weaknesses suggest that the paper may lack a comprehensive evaluation and may not provide substantial advancements over existing methods.

**Questions:**

See above

---

### Official Review · Reviewer_98Kb · 2023-10-31

**Soundness:** 3 good
**Presentation:** 3 good
**Contribution:** 2 fair
**Rating:** 3
**Confidence:** 4

**Summary:**

This paper proposes a T2I personalization approach based on low-rank residuals, which uses fewer parameters and does not rely on class regularization. A localized attention-guided sampling approach is proposed such that the learned concept only influences (the subject) parts of the image, preventing overfitting the learned concepts while generating subject-irrelevant content (like background).

**Strengths:**

The paper leverages LoRA to learn personalized concepts, despite being widely used by the community [1, 2], and discovers that such usages can preserve the existing diffusion prior, thus eliminating the need for class regularization. The paper also experiments with the optimal values of the LoRA rank, which may be useful to the community.

**Weaknesses:**

My primary concerns about the papers are two-fold.
1. Method-wise, the main contributions of this paper are to use LoRA for fewer trainable parameters and getting rid of regularization, and use attention maps to mask out the subject foreground. However, as mentioned in the strength part, using LoRA for training diffusion models is widely adopted in the communities [1], as well as for training personalized diffusion models [2]. Despite the paper exploring that leveraging LoRA preserves diffusion prior thus avoiding regularization, the reduction in trainable parameters is mainly contributed by the use of LoRA, thus the contribution is very incremental in my opinion. Secondly, previous works have also adopted cross-attention maps to guide the diffusion model to focus on the subject token such that the attention maps associated with each attended token will be activated [3]. More similarly, [4] leverages attention maps to suggest personalized models for foreground subject generation, which shares the same motivation and almost identical solution as this paper, while according to the time stamp it was released around half a year ago with their code available (June 1st, 2023 according to https://arxiv.org/abs/2306.00971).

2. Performance-wise, it is hard to distinguish the superiority of the proposed approach against DreamBooth. The comparison might be unfair qualitatively, however, as shown in Table 1, the proposed method does not lead to a compelling improvement against baselines like DreamBooth on CLIP text and image alignment metrics. Furthermore, as suggested in Table 2, more human evaluations indicate preferences over DreamBooth for better text alignment as well.
As above, from both the perspectives of presented methods and experimental evaluations, the paper does not show a clear distinction between the existing approaches.

[1] https://github.com/cloneofsimo/lora

[2] https://github.com/huggingface/peft/tree/main/examples/lora_dreambooth

[3] Chefer, Hila, et al. "Attend-and-excite: Attention-based semantic guidance for text-to-image diffusion models." ACM Transactions on Graphics (TOG) 42.4 (2023): 1-10.

[4] Hao, Shaozhe, et al. "ViCo: Detail-Preserving Visual Condition for Personalized Text-to-Image Generation." arXiv preprint arXiv:2306.00971 (2023).

**Questions:**

Please see the weaknesses above.

---

### Official Review · Reviewer_mRPT · 2023-11-01

**Soundness:** 3 good
**Presentation:** 3 good
**Contribution:** 4 excellent
**Rating:** 6
**Confidence:** 4

**Summary:**

This work addresses the challenge in concept-driven text-to-image generation motivated by a low-rank personalization approach.
The authors introduced a personalized residual computation for each transformer block.

They resolved the issue of concept forgetting in fine-tuning by leveraging their approach and also reduced the computational cost traditionally associated with fine-tuning a new set of parameters for each concept.

They proposed residual computations based on localized attention to enable fine-tuning with a small number of parameters, ensuring that the image adheres closely to the intended concept.

They utilized LAG (Localized Attention-Guided sampling) to improve the precision of image generation, particularly in personalization scenarios where specific local changes are required in the generated image.

The performance of their method showed competitive results against established baselines with fewer parameters and the user study showed a preference for their method.

**Strengths:**

The paper is overall well-written and easy to follow.

The authors proposed a novel concept-driven text-to-image generation technique, grounded in a low-rank personalization approach, that addresses challenges associated with traditional fine-tuning methods.

The paper demonstrates that localized attention-guided sampling effectively mitigates the overfitting to specific concepts. This design reflects a thoughtful integration of both attention and residuals. Additionally, a detailed analysis is presented, highlighting the advantages and limitations of LAG compared to conventional sampling technique.

The methods are robust and supported by empirical results. They introduce a unique solution to address the issue of concept forgetting in fine-tuning, making the approach both scalable and efficient.

**Weaknesses:**

Robustness: As the author mentioned, any shortcomings in the attention maps could significantly impair the overall performance of the model.

Macro class sensitivity: The choice of macro class can influence the performance of the model and its general applicability. Selecting the optimal macro class for certain datasets or domains, where the macro class is ambiguous, might require extensive fine-tuning or human trial-and-errors to select the right macro class.

Minor details: There is a typo in the introduction where “an” is misspelled as “am”.

**Questions:**

Given instances where the output of the model significantly diverges from the expected outcomes, can you propose or discuss any introspective mechanisms within the paper to determine which specific component or step might be responsible for the observed discrepancy?

---

### Official Review · Reviewer_dZ91 · 2023-11-01

**Soundness:** 2 fair
**Presentation:** 3 good
**Contribution:** 1 poor
**Rating:** 5
**Confidence:** 4

**Summary:**

In this paper, the authors proposed a method for subject-driven/personalized text-to-image generation. The proposed method is a combination of LoRA training for the output projection layers and constrains the effect of personalized weight on specific areas determined by the cross attention maps. The method shows comparable performance to competing baselines with relatively low computational requirements.

**Strengths:**

This paper is well written and very easy to follow. The qualitative examples shown in the paper demonstrate comparable performance against the baselines while providing some improvements on the training computation requirement and time.

**Weaknesses:**

1. My main concern about this paper is its novelty.

     (1) The paper can be summarized as “DreamBooth + LoRA + Paint-by-Words” and none of these components is new. The authors claim that their localized attention guidance (LAG) is a new method. However, the essence of this algorithm is to edit the cross attention maps using binary masks. This technique has been widely applied to many papers like Paint-by-Word and Prompt2Prompt which are two papers that the authors cited, and others like “Diffusion Self-Guidance for Controllable Image Generation” and “Localized Text-to-Image Generation for Free via Cross Attention Control” which the authors did not cite.

    (2) The authors also label the “DreamBooth + LoRA” method as “Ours”, but the only design choices that the authors made are the layers to apply LoRA and dropping the prior preservation loss. They also claim that the benefits from LoRA “eliminate the burden on the user to determine an effective set of regularization images”. However, DreamBooth uses ancestral sampling to generate the set of regularization images and it does not require the users to determine that. As a result, I don’t see the so-called “burden” being a very big problem.

2. The evaluation of the results does not contain any metrics for fidelity, which is a standard evaluation criterion adopted by most of the image generation tasks and the baselines the authors chose.

3. The authors did not mention any discussion about the limitations and failure case analysis of their method.

4. The quantitative improvement of the method is not very significant.

**Questions:**

It would be great if the authors can write out the learning objectives that they used in the paper